# Low Preoperative Antithrombin III Level Is Associated with Postoperative Acute Kidney Injury after Liver Transplantation

**DOI:** 10.3390/jpm11080716

**Published:** 2021-07-26

**Authors:** Kyoung-Sun Kim, Young-Jin Moon, Sung-Hoon Kim, Bomi Kim, In-Gu Jun, Hye-Mee Kwon, Jun-Gol Song, Gyu-Sam Hwang

**Affiliations:** Department of Anesthesiology and Pain Medicine, Laboratory for Cardiovascular Dynamics, Asan Medical Center, University of Ulsan College of Medicine, Seoul 05505, Korea; kyoungsun.kim@amc.seoul.kr (K.-S.K.); yjmoon@amc.seoul.kr (Y.-J.M.); shkimans@amc.seoul.kr (S.-H.K.); dudnd5@gmail.com (B.K.); igjun@amc.seoul.kr (I.-G.J.); hyemee.kwon@amc.seoul.kr (H.-M.K.); kshwang@amc.seoul.kr (G.-S.H.)

**Keywords:** acute kidney injury, antithrombin, liver transplantation

## Abstract

We aimed to determine the association between the preoperative antithrombin III (ATIII) level and postoperative acute kidney injury (AKI) after LT (post-LT AKI). We retrospectively evaluated 2395 LT recipients between 2010 and 2018 whose data of perioperative ATIII levels were available. Patients were divided into two groups based on the preoperative level of ATIII (ATIII < 50% vs. ATIII ≥ 50%). Multivariable regression analysis was performed to assess the risk factors for post-LT AKI. The mean preoperative ATIII levels were 30.2 ± 11.8% in the ATIII < 50% group and 67.2 ± 13.2% in the ATIII ≥ 50% group. The incidence of post-LT AKI was significantly lower in the ATIII ≥ 50% group compared to that in the ATIII < 50% group (54.7% vs. 75.5%, *p* < 0.001); odds ratio (OR, per 10% increase in ATIII level) 0.86, 95% confidence interval (CI) 0.81–0.92; *p* < 0.001. After a backward stepwise regression model, female sex, high body mass index, low albumin, deceased donor LT, longer duration of surgery, and high red blood cell transfusion remained significantly associated with post-LT AKI. A low preoperative ATIII level is associated with post-LT AKI, suggesting that preoperative ATIII might be a prognostic factor for predicting post-LT AKI.

## 1. Introduction

Acute kidney injury (AKI) after liver transplantation (LT) (post-LT AKI) is prevalent and a risk for morality [1,2]. Furthermore, AKI leads to the progression of chronic kidney injury [3]. Several perioperative factors, such as high model for end-stage liver disease (MELD) scores, pre-LT renal dysfunction, graft quality and intraoperative hemodynamic instability are attributed to the development of post-LT AKI [2,4,5]. Recently, inflammation was thought to be a potential mechanism for understanding of AKI [6,7].

Antithrombin III (ATIII), produced by the liver, exhibits properties of both anti-coagulation and anti-inflammation [8]. Previous studies regarding LT surgery only focused on the effect of ATIII as a serine protease inhibitor in the coagulation cascade to prevent hepatic artery thrombosis or portal vein thrombosis [9,10]. The reno-protective potential of ATIII has been consistently supported by evidence from numerous animal studies [11,12].

Patients with end-stage liver disease (ESLD) are known to have low levels of ATIII, which may result from impaired hepatic synthesis and increased consumption [13]. In addition, the systemic inflammation reaction and the associated immunological imbalance are proposed to integrate the main pathophysiological pathway of ESLD [6]. Moreover, the role of the inflammatory reaction has gained increasing recognition as a key factor for postoperative AKI [7,14,15].

In this regard, we hypothesized that the anti-inflammatory effect of ATIII might have a role in preventing AKI after LT surgery. This study aimed to describe the relationship between the ATIII level and post–LT AKI. In addition, we assessed the incidence of early allograft dysfunction (EAD), graft failure, chronic kidney disease (CKD), and overall mortality. 

## 2. Materials and Methods

### 2.1. Study Population

This retrospective observational study was approved by the institutional review board of our center (No.2019-1699). All patients who were reviewed underwent either living- or deceased-donor LT (LDLT, DDLT) from April 2010 to January 2018. Of the 3034 identified adults (≥18 years of age), re-transplantation cases (*n* = 85), patients who had been previously diagnosed with end-stage renal disease or CKD (*n* = 24), patients who were being treated with continuous renal replacement therapy (*n* = 411), and patients who did not have preoperative (*n* = 72) or postoperative (*n* = 47) ATIII levels were excluded. Finally, 2395 patients were included in our final analysis (Figure 1).

### 2.2. Perioperative ATIII Administration and Measurement

ATIII levels were measured preoperatively and postoperatively in all LT recipients according to our institution’s routine protocol. Preoperative ATIII values were measured at the day before transplant in all LT recipients. The study population was divided into two groups according to their preoperative ATIII values: preoperative ATIII ≥ 50% (the ATIII ≥ 50% group) vs. preoperative ATIII < 50% (the ATIII < 50% group). For all recipients, regardless of preoperative ATIII level, 2000 units of exogenous ATIII was administered daily for seven consecutive days, starting the day after the transplantation, according to our institution’s protocol. In addition, only for the recipients with baseline ATIII level lower than 50% (the ATIII < 50% group), exogenous ATIII (SK Anti-Thrombin III; SK plasma, Osan, South Korea) was administered during the anhepatic phase of the LT surgery according to our institution’s protocol. The administered dose was calculated as follows: Exogenous dose of ATIII (units)=[100−baseline ATIII value (%)]×body weight (kg)×0.8

### 2.3. Data Acquisition 

We collected patients’ baseline characteristics, laboratory variables, and perioperative variables using our institution’s medical record system. Baseline characteristics included age, sex, body mass index (BMI), the presence of an underlying disease (diabetes mellitus, hypertension, coronary artery disease), the model for end-stage liver disease (MELD), cause for liver transplant (hepatitis virus, alcoholics, malignancy, or others), and comorbidities due to ESLD (varix bleeding, hepatic encephalopathy, or pleural effusion). Preoperative laboratory variables included the level of ATIII, platelet count, total bilirubin, albumin, prothrombin time–international normalized ratio (INR), creatinine, and C-reactive protein. Perioperative variables consisted of donor type (deceased or living), operation time, graft-to-recipient weight ratio (GRWR), total ischemic time, amount of transfusion, and tacrolimus trough level within postoperative day 7. 

### 2.4. Postoperative Outcomes

The primary outcome was the occurrence of postoperative AKI, which was defined as positive when serum creatinine increases 0.3 mg/dL or more within postoperative day (POD) 2 or increases 1.5 times or more within POD 7, according to the creatinine criteria of the Kidney Disease: Improving Global Outcomes (KDIGO) classification [16]. It is further divided into three grades of acute kidney injury as follows: grade 1 is defined as an increase in creatinine ≥0.3 mg/dL within POD 2 or 1.5–1.9 times from baseline creatinine within POD 7, grade 2 is defined as an increase in creatinine 2.0–2.9 times from baseline creatinine within POD 7, grade 3 is defined as when an increase in creatinine is more than 3 times from baseline creatinine within POD 7, creatinine value is ≥4.0 mg/dL with an acute increase of at least 0.5 mg/dL within POD 7, or patients are treated with renal replacement therapy within POD 7. 

The secondary outcomes include progression to CKD, EAD, graft failure, and overall mortality. The occurrence of CKD was defined as positive when the glomerular filtration rate decreased to lower than 60 mL∙min^−1^∙1.73 m^−2^ on two consecutive occasions at least 3 months apart [17]. EAD was defined as positive when at least one of the following criteria was present: total bilirubin ≥10 mg/dL on POD7, prothrombin time (INR) ≥ 1.6 on POD 7, and alanine or aspartate aminotransferases >2000 IU/L within POD 7 [18]. 

### 2.5. Statistical Analysis

Values are expressed as numbers (percentages), mean ± standard deviation, or median (interquartile range (IQR)) according to the normality of the data. Analyses between groups were performed using the Student’s *t*-test, Mann–Whiney U test, analysis of variance, logistic regression, or Kruskal–Wallis test for continuous variables, and χ^2^ test or Fisher’s exact test for categorical variables, as appropriate. Significant variables with *p* value < 0.1 in the univariate analysis were entered into the multivariate logistic regression model using backward elimination. To investigate the prognostic value of the ATIII, subgroup analysis according to MELD scores (<20, ≥20) and donor types (deceased or living) were performed. All *p* values < 0.05 were considered statistically significant. All statistical analysis and graphical representations were performed using the R software version 3.3.2 (http://www.r-project.org) and Prism software version 7 (GraphPad, San Diego, CA, USA).

## 3. Results

### 3.1. Baseline Characteristics

Table 1 demonstrates the baseline characteristics of the patients according to the groups based on ATIII levels (<50%, ≥50%). The total cohort of 2395 patients had a mean age of 53.1 ± 8.5 years and a median (IQR) MELD score of 13 (9–18), while male patients comprised 74.7% of the study population. Causes for liver transplant were hepatitis-virus-related liver cirrhosis (67.4%), alcoholic liver cirrhosis (18.6%), and others (14.1%). The majority of the study population underwent living-donor LT (93.2%). The mean duration of surgery was 13.1 h, and 26.7% of the patients received a massive transfusion of 10 units of packed red blood cells (pRBCs) or more during the operation.

### 3.2. Perioperative AT III Value

In all, 15,884 perioperative ATIII values were measured from the day before transplant to POD 7. Preoperative ATIII values were measured for all 2395 patients. Postop-rative ATIII was measured at least once for each patient within POD 7. By POD 2, 97.1% of the total patients had at least one postoperative ATIII value measured. The mean preoperative ATIII value was 44.8 ± 21.9% (67.1 ± 13.2% in the ATIII ≥ 50% group vs. 30.2 ± 11.8% in the ATIII < 50% group, *p* < 0.001). The 1451 (60.6%) patients who had preoperative ATIII values lower than 50% (the ATIII < 50% group) received additional intraoperative exogenous ATIII in a dose calculated by our institution’s protocol. Postoperative ATIII levels showed an increasing trend in both groups. ATIII levels in the ATIII ≥ 50% group were higher than those in the AT < 50% group, except on POD 1, where the ATIII value of the ATIII < 50% group transiently exceeded that of the ATIII ≥ 50% group due to the additional intraoperative administration of recombinant ATIII (64.0 ± 18.2% for the ATIII ≥ 50% group vs. 79.0 ± 21.6% for the ATIII < 50% group, *p* < 0.001, Appendix A). 

### 3.3. Incidence of Postoperative Acute Kidney Injury and Its Relationship with ATIII Value

Postoperative AKI occurred in 67.3% of the total population. Grade 2 and 3 comprised 3.5% and 3.6% of total incidents of postoperative AKI, respectively (Table 2). Based on the preoperative ATIII value, the rate of postoperative AKI was significantly higher in the ATIII < 50% group than in the ATIII ≥ 50% group (75.5% vs. 54.7%, *p* < 0.001, Table 2). The probability of postoperative AKI decreased with the increasing value of preoperative ATIII (Figure 2A). 

In the multivariate analysis, preoperative ATIII remained as an independent predictor for postoperative AKI (adjusted odds ratio (OR, per 10% increase), 0.86; 95% confidence interval (CI) 0.81–0.92; *p* < 0.001) along with female sex (OR 1.26; 95% CI 1.01–1.57; *p* = 0.035), body mass index (OR 1.09; 95% CI 1.06–1.12; *p* < 0.001), serum albumin level (OR 0.62; 95% CI 0.53–0.74; *p* < 0.001), deceased-donor LT (OR 6.69; 95% CI 3.22–13.93; *p* < 0.001), duration of surgery (OR (per hour increase) 1.14; 95% CI 1.09–1.20; *p* < 0.001), and pRBC transfusion (OR (per 5 units increase) 1.08; 95% CI 1.02–1.14; *p* = 0.009) (Table 3). Figure 2B shows the plotted multivariate-adjusted relative risk of postoperative AKI in relation with preoperative ATIII level.

### 3.4. Development of Chronic Kidney Disease after Postoperative Acute Kidney Injury

CKD developed in 32.3%, 37.1%, and 35.8% of the patients after 3 months, 6 months, and 1 year of LT, respectively. Patients with ATIII < 50% are prone to postoperative CKD (*p* = 0.036, Table 2). AKI can lead to a higher prevalence of CKD at 3 months, 6 months, and 1 year after LT (all *p* < 0.001, Figure 3). Moreover, patients with ATIII < 50% showed a higher incidence of EAD (15.3% vs. 4.0%, *p* < 0.001). However, the overall graft failure and overall mortality after LT were not different according to the preoperative ATIII level (Table 2).

### 3.5. Subgroup Analysis According to the MELD Scores and Donor Types

The patients were further divided into subgroups based on MELD scores (≥20, <20) and donor types (deceased vs. living) to evaluate the predictive value of the ATIII. In the subgroups divided by MELD score, ATIII had a preventive effect on AKI in patients with MELD scores < 20 (OR 0.82, 95% CI 0.77–0.87, *p* < 0.001, Supplement Appendix A). In contrast, ATIII did not have a statistically significant preventive effect in patients with MELD scores ≥ 20. In patients who underwent DDLT, the median MELD score was 26, while it was 12 in patients who underwent LDLT. The ATIII level was 22% in the DDLT subgroup and 45% in the LDLT subgroup. The AKI incidence was higher in DDLT (86.6% vs. 65.8%, *p* < 0.001). When analyzing the ATIII value by donor types (deceased vs. living), the predictive value of ATIII for AKI in LDLT was statistically significant (OR 0.89, CI 0.85–0.94, *p* < 0.001).

## 4. Discussion

In this retrospective observational study, we found that the preoperative level of ATIII was proportionally associated with postoperative AKI in over 2300 patients undergoing LT. In multivariate analysis, a 10% increase in preoperative ATIII was associated a 0.86-fold decreased risk of post-LT AKI. This association between preoperative ATIII level and post-LT AKI remained significant only in the low-MELD group (<20) and LDLT. Additionally, low ATIII was associated with the progression of CKD and EAD.

In studies regarding the relationship between ATIII and AKI, as the majority of previous studies have been performed in animal models, the human data are limited in their number [11,12,19,20]. A previous study by Wang et al. demonstrated the predictive value of the ATIII level in the development of AKI [11]. The authors concluded that ATIII appears to ameliorate renal ischemia–reperfusion injury through inhibiting inflammatory response, oxidative stress, apoptosis, and by improving renal blood flow. In this study, the authors also presented human clinical data, which included only seven patients with low ATIII levels (<75%) and the majority of the patients had a normal range of ATIII levels. Another recent study by Park et al. demonstrated the relationship between preoperative low ATIII levels (<70%) and AKI following LDLT [21]. The authors suggested that the incidence of AKI was 24.8% (143/577) following LDLT, which was much lower than our study (67.3%, 1611/2395). We assumed that the difference of AKI incidence might be due to the study population. In our study, we included DDLT recipients, who are a high-risk group for the development of post-LT AKI. As the patients’ severity increased, the operation time was about 4 h longer than previous study [21]. Consistent with our results, a previous study showed that operation time >10 h was an independent risk factor for post-LT AKI [22]. In addition, our results showed that the massive transfusion rate (10 > units of pRBC) was 26.7%, which is one of the risk factors of AKI. 

Our center’s protocol of ATIII administration during perioperative period resulted in a rapid increase in ATIII level. We uniformly administered 2000 units of ATIII daily from POD1 to POD7. Kaneko et al. reported that the preoperative ATIII levels did not return to normal after LDLT during the first 2 weeks [23]. Previously, an initial ATIII level < 50% was reported as the best prognostic value for the prediction of mortality from septic shock [24]. Therefore, only the patients who had preoperative an ATIII level < 50% were given an additional dose of ATIII during the surgery. As a consequence, the average level of ATIII reached a normal level from POD 2 (88.6 ± 18.4% for ATIII ≥ 50% group and 85.9 ± 17.5% for AT III < 50% group). However, at POD7, the ATIII level exceeded the normal level (135.1 ± 24.1% for ATIII ≥ 50% group and 124.2 ± 21.3% for ATIII < 50% group). A protocol with individualized dosage according to the ATIII level might be more cost effective for maintaining a consistent level of ATIII during the post-LT period. 

In our study, low ATIII level, female gender, high BMI, low serum albumin, DDLT, longer operation time, and more pRBC transfusions were considered risk factor for development of post-LT AKI. Similarly, previous study also reported that female gender, obesity, low serum albumin, DDLT, longer operation time, and more blood transfusions were associated with the development of post-LT AKI [1,2,22,25]. In our study, MELD score was not a risk factor for post-LT AKI. As we excluded patients who were previously diagnosed with end-stage renal disease or CKD, the MELD score did not significantly impact the incidence of post-LT AKI. According to our results, preoperative ATIII as a prognostic value for predicting AKI remained significant only in the low-MELD group (<20) and LDLT, but was not statistically significant in the high-MELD group (≥20) and DDLT. Given that the incidence of AKI in the DDLT group was over 80% in our study, the DDLT group may be affected by multifactorial variables that might have influenced the post-LT AKI. In other studies, consistent with our result, the DDLT group had a higher incidence of post-LT AKI than the LDLT group [25].

In our study, more than half of the patients developed post-LT AKI and a significant number of these cases led to CKD after LT surgery. Although most of them were grade I AKI (89.5%, 1442/1611), over one third of them developed into CKD (32.3% at 3 months, 37.1% at 6 months, 35.8% at 1 year after LT). Given that both AKI and CKD developed in LT recipients are known as risk factors of the utmost importance for post-LT morbidity and mortality, [26] a prevention strategy of AKI should be implemented in the peri-LT period. In addition, considering the relatively high rate of postoperative AKI in LT surgery compared to other non-cardiac surgeries, there is more room for improvement [27]. 

There are several limitations presented in this study. First, we could not explain the mechanism of ATIII on the development of post-LT AKI. However, a previous study demonstrated that ATIII insufficiency exacerbates renal ischemia–reperfusion injury by inflammation, oxidative stress, and apoptosis, [11] which are some of the well-known mechanisms associated with the development of AKI. Second, as this study was designed retrospectively, the confounding factors could not be eliminated entirely. Third, there is no clear evidence regarding the perioperative optimal dose of ATIII administration to prevent AKI in LT. Although the ATIII level could be normalized during first two weeks after LT, [23] to mitigate post-LT AKI incidence, it is necessary to find the target level of ATIII. Therefore, further study is needed for the individualized administration dose based on preoperative ATIII level.

## 5. Conclusions

In conclusion, we found that a low level of ATIII is significantly associated with post-LT AKI. Notably, preoperative ATIII might have prognostic value for post-LT AKI in the low-MELD group (<20) and LDLT in our study. 

## Figures and Tables

**Figure 1 jpm-11-00716-f001:**
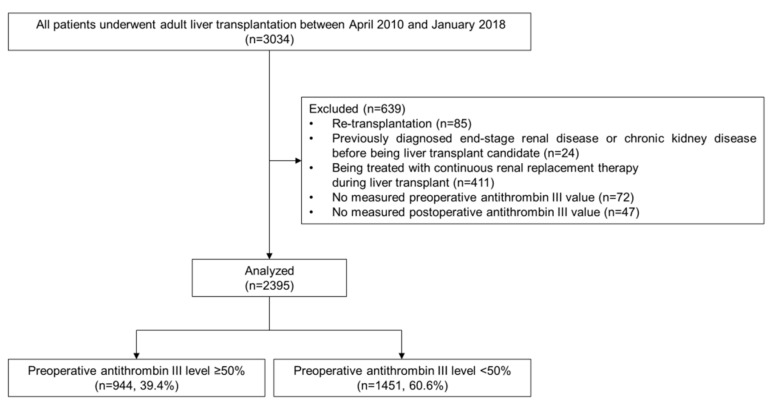
Flow diagram of the patient inclusion and exclusion.

**Figure 2 jpm-11-00716-f002:**
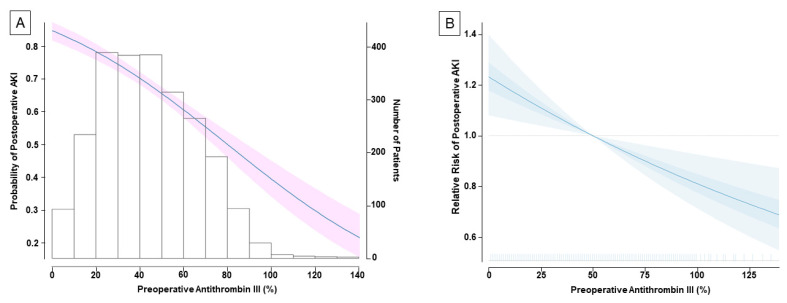
The association between preoperative antithrombin III (ATIII) and postoperative acute kidney injury. The solid line indicates the probability on a continuous scale value, and the shaded areas indicate 95% confidence interval (**A**). The multivariate-adjusted relative risk plot showing the relationship between preoperative ATIII and postoperative acute kidney injury (**B**). Estimates are adjusted for independent confounders from multivariable generalized logistic regression model. The solid lines and translucent band depict relative risk and 95% confidence intervals of those estimates.

**Figure 3 jpm-11-00716-f003:**
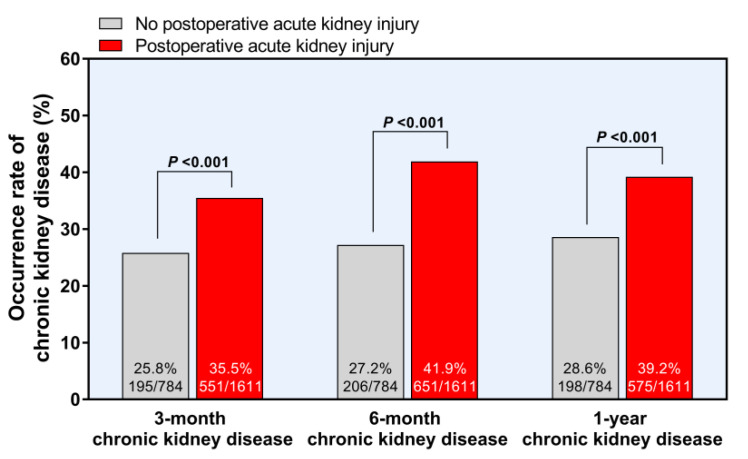
Development of chronic kidney disease after postoperative acute kidney injury.

**Table 1 jpm-11-00716-t001:** Patients’ demographics according to the preoperative antithrombin III level.

	Antithrombin III < 50% (N = 1451, 60.6%)	Antithrombin III ≥ 50 % (N = 944, 39.4%)	Total (N = 2395)	*p* Value
**Demographic data**				
Age (years)	52.8 ± 8.6	53.5 ± 8.2	53.1 ± 8.5	0.036
Sex (male sex)	1059 (73.0%)	730 (77.3%)	1789 (74.7%)	0.019
Body mass index (m/kg^2^)	24.0 ± 3.7	23.8 ± 3.2	23.9 ± 3.5	0.304
Diabetes	350 (24.1%)	207 (21.9%)	557 (23.3%)	0.233
Hypertension	191 (13.2%)	203 (21.5%)	394 (16.5%)	<0.001
Coronary artery disease	13 (0.9%)	13 (1.4%)	26 (1.1%)	0.363
MELD score	16 (12–23)	8.5 (7–11)	13 (9–18)	<0.001
MELD score over 20	462 (31.8%)	19 (2.0%)	481 (20.1%)	<0.001
**Causes for liver transplant**				<0.001
HBV-related liver cirrhosis	526 (67.1%)	922 (57.2%)	1448 (60.5%)	
HCV-related liver cirrhosis	42 (5.4%)	123 (7.6%)	165 (6.9%)	
Alcoholic liver cirrhosis	118 (15.1%)	327 (20.3%)	445 (18.6%)	
Others	98 (12.5%)	239 (14.8%)	337 (14.1%)	
Hepatocellular carcinoma	452 (57.7%)	782 (48.5%)	1234 (51.5%)	
**Comorbidities**				
Varix bleeding	386 (26.6%)	248 (26.3%)	634 (26.5%)	0.895
Hepatic encephalopathy	273 (18.8%)	22 (2.3%)	295 (12.3%)	<0.001
Pleural effusion	252 (17.4%)	51 (5.4%)	303 (12.7%)	<0.001
**Preoperative laboratory variables**				
Antithrombin III (%)	30.2 ± 11.8	67.1 ± 13.2	44.8 ± 21.9	<0.001
Platelet count (×10^3^/µL)	65.3 ± 43.1	90.3 ± 58.4	75.1 ± 51.2	<0.001
Total bilirubin (mg/dL)	8.3 ± 10.5	1.8 ± 4.0	5.7 ± 9.1	<0.001
Albumin (g/dL)	3.0 ± 0.6	3.4 ± 0.5	3.1 ± 0.6	<0.001
Prothrombin time (INR)	1.9 ± 1.2	1.2 ± 0.2	1.6 ± 1.0	<0.001
Creatinine (mg/dL)	0.8 ± 0.5	0.8 ± 0.3	0.8 ± 0.4	0.502
C-reactive protein (mg/L)	0.8 ± 1.1	0.4 ± 0.9	0.7 ± 1.1	<0.001
**Operative variables**				
Deceased donor	149 (10.3%)	15 (1.6%)	164 (6.8%)	<0.001
Living donor	762 (97.2%)	1469 (91.2%)	2231 (93.2%)	0.001
Duration of surgery (min)	800.2 ± 151.0	758.0 ± 135.2	783.6 ± 146.4	<0.001
Graft-to-recipient weight ratio (g/kg)	1.2 ± 0.5	1.1 ± 0.3	1.2 ± 0.4	<0.001
Total ischemic time (min)	147.5 ± 71.7	130.5 ± 53.4	140.8 ± 65.6	<0.001
Transfusion				
pRBC (unit)	10.4 ± 14.7	3.4 ± 6.8	7.7 ± 12.7	<0.001
* Massive transfusion	535 (36.9%)	104 (11.0%)	639 (26.7%)	<0.001
FFP (unit)	10.8 ± 14.9	3.5 ± 7.0	7.9 ± 12.9	<0.001
Cryoprecipitate (unit)	8.1 ± 8.9	2.8 ± 5.2	6.0 ± 8.1	<0.001
Platelet apheresis (unit)	0.8 ± 1.0	0.3 ± 0.6	0.6 ± 0.9	<0.001
Tacrolimus trough level within POD 7	5.2 ± 2.6	6.0 ± 2.5	5.5 ± 2.6	<0.001

Continuous variables are expressed as mean ± standard deviation or as median (interquartile range) and categorical variables as n (%). * Massive transfusion: transfusion of >10 units of packed red blood cells. MELD score, model for end-stage liver disease score; IQR, interquartile range; INR, international normalized ratio; pRBC, packed red blood cells; FFP, fresh frozen plasma; POD, postoperative day.

**Table 2 jpm-11-00716-t002:** Postoperative outcomes according to preoperative antithrombin III levels.

	Preoperative Antithrombin III < 50% (N = 1451)	Preoperative Anithrombin III ≥ 50% (N = 944)	Total (N = 2395)	*p* Value
**Postoperative renal outcomes**				
Acute kidney injury	1095 (75.5%)	516 (54.7%)	1611 (67.3%)	<0.001
Grades of acute kidney injury				<0.001
1	970 (66.9%)	472 (50.0%)	1442 (60.2%)	<0.001
2	49 (3.4%)	34 (3.6%)	83 (3.5%)	0.858
3	76 (5.2%)	10 (1.1%)	86 (3.6%)	<0.001
Chronic kidney disease				
At 3 months after liver transplant	474 (34.1%)	272 (29.6%)	746 (32.3%)	0.029
At 6 months after liver transplant	551 (39.6%)	306 (33.3%)	857 (37.1%)	0.003
At 1 year after liver transplant	490 (37.6%)	283 (33.1%)	773 (35.8%)	0.036
**Other postoperative outcomes**				
Early allograft dysfunction	222 (15.3%)	38 (4.0%)	260 (10.9%)	<0.001
Overall graft failure	115 (7.9%)	67 (7.1%)	182 (7.6%)	0.504
Overall mortality	109 (7.5%)	63 (6.7%)	172 (7.2%)	0.487

**Table 3 jpm-11-00716-t003:** Multivariate analysis of risk factors associated with postoperative acute kidney injury.

	All Patients (N = 2395)			
	Crude OR (95% CI)	*p* Value	Multivariate OR (95% CI)	*p* Value
Preoperative antithrombin III (per 10% increase)	0.81 (0.78–0.84)	<0.001	0.86 (0.81–0.92)	<0.001
Age	1.00 (1.00–1.01)	0.257		
Female sex	1.23 (1.01–1.51)	0.041	1.26 (1.01–1.57)	0.035
Body mass index	1.09 (1.07–1.12)	<0.001	1.09 (1.06–1.12)	<0.001
Diabetes mellitus	1.30 (1.06–1.61)	0.012		
Hypertension	1.22 (0.96–1.54)	0.101		
Hepatic encephalopathy	1.66 (1.25–2.20)	<0.001		
MELD score	1.04 (1.03–1.06)	<0.001		
Albumin	0.50 (0.43–0.58)	<0.001	0.62 (0.53–0.74)	<0.001
C-reactive protein	1.15 (1.05–1.26)	0.003		
Deceased-donor LT	3.35 (2.12–5.29)	<0.001	6.69 (3.22–13.93)	<0.001
Operation time (per hour increase)	1.14 (1.10–1.19)	<0.001	1.14 (1.09–1.20)	<0.001
pRBC transfusion(per 5 units)	1.20 (1.14–1.27)	<0.001	1.08 (1.02–1.14)	0.009
Graft-to-recipient weight ratio	1.21 (0.98–1.49)	0.082		
Total ischemic time (per 10 min)	1.04 (1.02–1.05)	<0.001		

OR, odds ratio; CI, confidence interval; LT: liver transplantation; MELD score, model for end-stage liver disease score; pRBC, packed red blood cells.

## Data Availability

The datasets used and analyzed during the current study are available from the corresponding author upon reasonable request.

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
