# Peer review of "Low Preoperative Antithrombin III Level Is Associated with Postoperative Acute Kidney Injury after Liver Transplantation"

_jpm, 2021, doi:10.3390/jpm11080716_

Round 1
Reviewer 1 Report
This author concluded that preoperative lower AT3 lever would be associated with postoperative acute kidney injury (AKI) strongly. This study was so interesting from our clinical point of view.
- If we give the patient (AT3 <50%) AT 3 administration preoperatively, can we prevent postoperative AKI? What do you think about the mechanism which lower AT3 level cause AKI?
- Why did not they include MELD score into multivariate analysis for AKI?
- What do you think about the reason why lower AT 3 level would cause AKI 1 and 3 not 2?
Author Response
"Please see the attachment

Reviewer 2 Report
REVIEW
„Low preoperative antithrombin III level is associated with 2 postoperative acute kidney injury after liver transplantation”
The authors of the current article aimed to determine the association between preoperative Antithrombin III (ATIII) 11 level and postoperative acute kidney injury (AKI) after LT (post-LT AKI).
The issue of the current article is extremely interesting and there ar emany patients suffer from renal diseases leading to kidney failure. More and more studies are needed to be performer to improve organ transplantation, including renal one.
All of the chapters were properly desribed. Authors very carefully performer their studies and they described the methods very clearly. They supported this chapter additionally with Figure which makes their article more clear.
Tables and Figures in Result sections are also clear.
Discussion is performed correctly and as far as I am concerened it is interesting.
The authors end the manuscipt with the proper conclusion- „Low preoperative ATIII level is associated with post-LT AKI, suggesting that preoperative 22 ATIII might be a prognostic factor for predicting post-LT AKI”.
I hope that it is going to be next step for improving the medicine in transplantology field.
I recommend to publish the article, however I advise to form the stronger conclusion with more details.
